# Vascular Endothelial Growth Factor A VEGFA Inhibition: An Effective Treatment Strategy for Psoriasis

**DOI:** 10.3390/ijms25010059

**Published:** 2023-12-20

**Authors:** Ya Chen, Zongguang Tai, Congcong Zhu, Qin Yu, Quangang Zhu, Zhongjian Chen

**Affiliations:** Shanghai Skin Disease Hospital, School of Medicine, Tongji University, 1278 Baode Road, Shanghai 200443, China; c2870404460@163.com (Y.C.); taizongguang@126.com (Z.T.); ccongzhu@126.com (C.Z.); shiyanyq@126.com (Q.Y.)

**Keywords:** vascular endothelial growth factor A, psoriasis, angiogenesis, epithelial immune microenvironment, therapeutic target

## Abstract

Psoriasis is an inflammatory skin disease mediated by the immune system and characterized by an inflammatory ring, also known as an epithelial immune microenvironment (EIME). The interaction between the epithelial tissue of the skin and the immune system has a crucial role in the immune cycle of psoriasis. Although the formation of new blood vessels in skin lesions provides energy support for the proliferation of epidermal keratinocytes, the role of angiogenesis in psoriasis has not been extensively studied. Vascular endothelial growth factor A (VEGFA) is a key regulator of angiogenesis that has an important role in the development of psoriasis. VEGFA promotes angiogenesis and directly stimulates epidermal keratinocytes and infiltrating immune cells, thus contributing to the progression of psoriasis. Measuring VEGFA levels to identify angiogenic characteristics in psoriasis patients may be a predictive biomarker for disease severity and response to anti-angiogenic therapy. Clinical data have shown that anti-angiogenic therapy can improve skin lesions in psoriasis patients. Therefore, this study aimed to uncover the underestimated role of blood vessels in psoriasis, explore the relationship between VEGFA and keratinocytes in the EIME, and inspire innovative drug therapies for the treatment of psoriasis.

## 1. Introduction

Psoriasis is a chronic inflammatory skin disease that affects 1% to 3% of the global population [1]. It is characterized by hyperproliferation of keratinocytes, abnormal vascular proliferation, and infiltration of immune factors [2]. The skin, the body’s first line of defense, can protect the body and repair damage. The structure of human skin has three parts, i.e., the epidermis, dermis (including the papillary dermis and reticular dermis), and subcutaneous connective tissue. All three parts contain specific non-immune and immune cells [3] (Figure 1). The epithelial immune microenvironment (EIME) is formed at the shallowest 0.1 to 0.2 mm of the epidermis and papillary dermis, and it dominates the immune response of many pathological skin diseases, including psoriasis. Neutrophils are the main and unique participants in the EIME of psoriasis, and their infiltration of the epidermis is an important histological and immunological marker of psoriasis [4].

The formation of papillary dermal microvessels is one of the necessary pathological features of psoriasis. So far, most research on psoriasis has focused on the immune components of the disease, while the role of blood vessels in the onset and maintenance of psoriasis has received little attention [5]. Although psoriasis is primarily a T-cell-driven disease, its pathophysiology is highly regulated by abnormalities in the papillary dermal vascular system. The keratin 14 vascular endothelial growth factor (K14-VEGF) transgenic animal model can spontaneously produce psoriasis phenotype characteristics [6]. Compared with non-lesional skin, vascular endothelial growth factor A (VEGFA) and its receptors vascular endothelial growth factor receptor-1 (VEGFR-1)/Fms-related receptor tyrosine kinase 1 (Flt1) and vascular endothelial growth factor receptor-2 (VEGFR-2)/fetal liver kinase 1 (Flk1) are increased in psoriatic plaques [7,8,9]. Various cell types express VEGFR-1/Flt1, while VEGFR-2/Flk1 is specifically expressed by endothelial cells [10]. The VEGFA levels in the plasma of patients with psoriasis are also increased, which is related to the severity of the disease [11]. The possible mechanisms of action mainly manifest due to the fact that VEGFA can directly promote the development of psoriasis by stimulating keratinocytes and infiltrating immune cells, promoting angiogenesis by promoting endothelial cell proliferation, which enhances autologous transcriptional activity in conjunction with hypoxia-inducible factor-1α (HIF-1α). Also, VEGFA regulates abnormal keratin expression. Therefore, VEGFA has a key role in the pathogenesis of psoriasis. Some anti-VEGFA small molecule inhibitors, such as bevacizumab [12], can effectively improve psoriatic lesions, but the specific mechanisms of action need further study.

This study aimed to investigate the role of VEGFA in the EIME of psoriasis. Our findings support the view that the VEGFA and its receptor signaling pathway might be a promising target for future psoriasis treatment interventions, and inhibiting angiogenesis is one of the effective treatment strategies for psoriasis [13]. This treatment strategy expands and optimizes the options for future individualized treatment of psoriasis [14,15].

## 2. Angiogenesis in the Pathogenesis of Psoriasis

Angiogenesis is a complex and highly ordered process that relies on an extensive signaling network of endothelial cells, vascular smooth muscle cells (VSMCs), pericytes, and other cell types, such as immune cells [16]. For instance, endothelial cells and macrophages synthesize and release many pro-angiogenic factors that promote angiogenesis in various ways after changing the local microenvironment under different stimuli, such as VEGF, angiogenin, platelet-derived growth factor (PDGF), transforming growth factor-β (TGF-β), basic fibroblast growth factor (bFGF), yes-associated protein 1 (YAP1), and PDZ-binding motif transcriptional coactivator (TAZ) [17]. Auspitz (punctate bleeding) is a characteristic of plaque psoriasis and a clinical manifestation of papillary dermal microvessel formation [18]. Angiogenesis in psoriatic plaques typically does not involve the growth of new capillaries from the original vessels in a classical sense but is characterized by significant vascular expansion and increased vascular permeability, i.e., inflammatory angiogenesis or pathological angiogenesis. In the context of psoriasis, the interactions among inflammatory factors, endothelial cells, and fibroblasts and the fact that the same molecular events can trigger inflammatory factors and angiogenesis further confirm their interrelationship [13,14,19,20].

In 1972, Folkman proposed that the earliest pathological changes in psoriatic lesions are abnormal distribution and formation of vessels [21]. Newly formed vessels provide nutrients for hyperproliferative keratinocytes and promote the migration of inflammatory cells. The development of new vessels is the third critical link in the pathogenesis of psoriasis, crucial for the onset and persistence of psoriasis. The improvement of psoriatic lesions is accompanied by normalizing vascular structures. In the inflammatory environment of psoriasis, a large number of pro-angiogenic factors are activated, the function of vascular endothelial cells changes, and endothelial cells induce adhesion molecules and promote leukocyte recruitment. In addition, regulatory T cells influence the angiogenic microenvironment by regulating related vascular endothelial growth factors, further promoting the proliferation of epidermal vessels. In conclusion, angiogenesis has a crucial role in psoriasis.

## 3. VEGFA Is the Most Important Vascular Endothelial Growth Factor

The mechanism of angiogenesis is strictly regulated within the body. Under pathological conditions, the balance between pro-angiogenesis and anti-angiogenesis is disrupted, leading to excessive vascular proliferation. This process is primarily mediated by VEGF, i.e., a member of the pro-angiogenic factor family with a molecular weight of 40–45 kDa, the strongest stimulator of angiogenesis. VEGF was discovered in 1986 [22] and first isolated in 1989 [23]. Its mechanism of action in the angiogenesis process was also discovered [24]. The tertiary structure of the VEGF protein is two reverse parallel homodimers, and its family members include VEGFA, VEGFB, VEGFC, VEGFD, VEGFE, and placental growth factor (PlGF) [25] (Figure 2). VEGFA has a crucial role in regulating normal and disease-related angiogenesis. Unless otherwise specified, most studies use VEGF referring to the VEGFA subtype. VEGFA protein is a glycosylated mitogen that specifically acts on endothelial cells and exerts multiple functions, including mediating increased vascular permeability, inducing angiogenesis and endothelial cell growth, promoting cell migration, and inhibiting cell apoptosis [26]. VEGFA is significantly associated with vascular endothelial cells, mainly due to its binding with receptors on the endothelial cell membrane, activating downstream pathways. At the same time, it also significantly impacts many other cell types, such as monocytes/macrophages [27], neurons [28], cancer cells [29], and renal epithelial cells [30]. VEGFA is also a vasodilator that can increase microvascular permeability and was initially identified as a vascular permeability factor.

The *VEGFA* gene can be edited to produce several splice isoforms, including VEGF121, VEGF145, VEGF162, VEGF165, VEGF183, VEGF189, and VEGF206 [31,32,33,34]. These *VEGFA* subtypes have different affinities for heparan sulfate proteoglycan (HSPG). The balance between freely diffusing and HSPG-bound VEGFA creates a gradient, forming a new tip cell, i.e., an endothelial cell responsive to angiogenic signal transduction. Tip cells lead the formation of angiogenic sprouts and ultimately promote vascular branching through various cell migration steps. With further vascular maturation and hemodynamic changes, endothelial cells secrete platelet-derived growth factor-B (PDGF-B), recruiting pericytes and VSMCs. These mural cells stabilize the growing vascular bed by binding to endothelial cells through angiopoietin-1 expression, thus resulting in TGF-β activation and extracellular matrix deposition. Downstream effectors, including phosphoinositide 3′-kinase (PI3K) [35], Src kinase [36], focal adhesion kinase (FAK) [37], p38/mitogen-activated protein kinase (p38/MAPK) [38], Smad2/3 [39], and phospholipase Cγ (PLC-γ)/(ERK1/2) [40] promote endothelial cell survival, vascular permeability, and migratory/proliferative phenotypes. These subtypes are expressed at different levels in different tissues, and each subtype can bind to different receptors, facilitating the diverse biological needs of angiogenesis in various tissues.

## 4. VEGFA Regulates the EIME in Psoriasis

There is sufficient evidence from preclinical to clinical studies confirming the importance of VEGFA in psoriasis (Table 1). The role of VEGFA in psoriasis can be considered from two dimensions. On the one hand, the genomic location of the *VEGFA* gene is near psoriasis susceptibility 1 (PSORS1). The *VEGFA* gene, which encodes VEGFA, is located at band 1, sub-band 3, of the short arm region 2 of chromosome 6 (6p21.3). On the other hand, there is a correlation between VEGFA single-nucleotide polymorphisms (SNPs) and the severity of psoriasis. Young and others provided genetic evidence for VEGFA as a modifier gene in psoriasis by analyzing VEGFA gene SNPs in psoriasis patients and healthy individuals [41,42]. Overexpression of VFGFA in mouse epidermis can induce psoriasis-like disease, and angiogenesis and inflammatory cell infiltration are similar to psoriatic skin lesions. Conditional loss of VEGFR-1/Flt1 or neuropilin-1 (Nrp1) in epidermal cells can reverse the psoriasis-like phenotype of VEGFA overexpressing mice, suggesting the role of blocking the VEGFA/Flt1/Nrp1 axis in psoriasis [43]. In psoriatic skin, VEGFA is mainly produced by activated keratinocytes, with a small amount of VEGFA produced by fibroblasts [44] and immune mediators such as hypertrophic cells [45]. The level of VEGFA produced by keratinocytes is significantly increased in psoriatic plaques [46] and is related to clinical severity [47]. In summary, studies on psoriasis mouse models and psoriasis patients have confirmed the critical role of VEGFA in psoriasis.

### 4.1. VEGFA Promotes Keratinocyte Proliferation

One of the clinical phenotypes of psoriasis is the proliferation of keratinocytes, and overexpression of VEGFA can promote the proliferation of basal keratinocytes. In psoriasis, keratinocytes are considered the main targets of effector cell cytokines such as interleukin (IL)-17 and IL-22; however, they do not participate in the pathogenesis of psoriasis. However, some familial psoriasis cases suggest abnormal cell signaling and transcription responses in keratinocytes may lead to psoriasis in genetically susceptible populations, and keratinocytes produce neutrophil chemotactic factors in psoriasis inflammation, such as chemokine (C-X-C motif) ligand 1 (CXCL1) and chemokine (C-X-C motif) ligand 2 (CXCL2) [57]. Similarly, studies have shown that keratinocytes are the target cell types of Th17-induced psoriasis skin inflammation. IL-17 activates epidermal keratinocytes through IL-17R, producing pro-inflammatory cytokines and chemokines, such as IL-1, IL-6, CXCL1, and CCL20, which are known to induce and propagate psoriasis inflammation [58]. The high proliferation state of keratinocytes in psoriasis patients may not only be related to VEGFA stimulating local angiogenesis and increased permeability through paracrine effects to obtain more nutrients but may also be related to VEGFA directly promoting keratinocyte proliferation through autocrine effects. Using exogenous VEGFA165 as a stimulant to stimulate keratinocytes while using a VEGFA inhibitor to stimulate cells showed that VEGFA165 can dose-dependently promote cell proliferation and migration, and after using the inhibitor, proliferation and migration were inhibited. The fact that keratinocytes in psoriatic lesions can express and secrete more VEGFA further verifies that VEGFA directly promotes keratinocyte proliferation through autocrine effects. Further studies on its mechanism found that basal keratinocytes express VEGFR-1/Flt1 and Nrp1, unlike VEGFR-2/Flk1. The absence of VEGFR-1/Flt1 in the epidermis completely prevented the proliferation of keratinocytes after VEGFA overexpression and the increase in skin immune infiltration. The absence of Nrp1 in the epidermis also prevented the occurrence of microscopic changes characteristic of psoriatic skin, including epidermal thickening, basal keratinocyte proliferation, immune infiltration, and increased microvascular density associated with VEGFA overexpression, indicating that Flt1 and Nrp1 interact in an upregulated manner genetically to promote VEGFA-induced keratinocytes proliferation [52].

### 4.2. VEGFA Promotes Keratin Expression in Psoriatic Epithelium

As keratinocytes are located at the interface between the body and the environment, they can receive signals from both the environment and other cells. They interact with immune cells in the skin, which is facilitated through the expression of receptors that perceive microbes and other environmental stressors, the production of cytokines and chemokines (including IL-1, IL-10, IL-20, and members of the TNF cytokine family), and the expression of cytokine and chemokine receptors. Many pro-inflammatory signaling cascades are activated in the epidermis of patients with inflammatory skin diseases [59,60,61].

Keratin, one of the main structural proteins of keratinocytes, forms intermediate filaments in the cytoskeleton and participates in epidermal differentiation and keratinization [1]. As a member of the fibrous structural protein family, keratin is the main protein of the outer layer of human skin, which protects epithelial tissues from damage. Abnormal keratin expression in psoriasis is critical in keratinocyte dysfunction and disease progression and is considered a disease hallmark [62]. In normal skin, K1 and K10 are expressed in the suprabasal (granulos) layer, while K5 and K14 are expressed in the basal layer [63,64]. K6, K16, and K17 are typically expressed in the palmar and plantar regions and other skin appendages. However, in cases of psoriasis or hyperplasia of keratinocytes, they can also be expressed in the basal upper layer of the epidermis. K6 and K10 serve as markers for keratinocyte proliferation and differentiation, respectively. They may be expressed individually or co-expressed in cells above the basal layer of the epidermis. K16 regulates the innate immunity of epithelial cells [65] and the activation of nuclear factor erythroid 2-related factor 2 (Nrf2) [66]. K17 appears to trigger inflammatory responses by stimulating the proliferation of psoriatic T cells and the release of interferon (IFN)-γ. It even regulates the transcription of K16, whose overexpression can promote the proliferation of keratinocytes [67]. As an autoantigen in psoriasis, K17 can stimulate T-cell proliferation and IFN-γ production [68]. VEGFA can upregulate the mRNA and protein expression of K6, K16, and K17 in normal human epidermal keratinocytes and downregulate the mRNA and protein expression of K1 and K10. This phenomenon may be related to activating the signal transducer and activator of transcription 3 (STAT3), extracellular regulated protein kinases 1/2 (ERK1/2), and p38 pathways in keratinocytes [69].

### 4.3. VEGFA Regulates the Immune Response in EIME

VEGFA is implicated in the aberrant expression of keratin in psoriatic skin tissue and the proliferation of keratinocytes. These keratinocytes produce VEGFA under the influence of immune factors, thereby modulating the proliferation of endothelial cells and human dermal microvascular endothelial cells (HDMECs). Consequently, this regulatory mechanism plays a crucial role in the EIME of psoriasis [70]. VEGF121 and VEGF165 are the main VEGFA subtypes found in the affected and non-affected skin of psoriasis patients [71,72]. IL-17 is a pro-inflammatory factor secreted by CD4^+^ T cells, which can induce epithelial cells, endothelial cells, and fibroblasts to synthesize and secrete IL-6, IL-8, granulocyte colony-stimulating factor, prostaglandin E2 (PGE2), and promote the expression of intercellular adhesion molecule-1 (ICAM-1). Tumor necrosis factor-α (TNF-α) is a pro-inflammatory cytokine mainly produced by macrophages and monocytes, involved in normal inflammation and immune responses. In normal tissues, keratinocytes produce VEGFA, promoting normal angiogenesis. In the pathogenesis of psoriasis, IL-17A and TNF-α jointly stimulate skin keratinocytes to release VEGFA, leading to pathological angiogenesis [73,74,75]. Some studies have shown that IL-17A and IL-22 can jointly stimulate skin keratinocytes to release VEGFA. IL-22 is a member of the IL-10 cytokine family, which can promote the proliferation, remodeling, and repair of various tissues and organs; maintain the inherent defense mechanisms of the host; and control pathogen invasion. DCs recognize pathogens in the innate immune system and activate immune cells in the adaptive immune system. Plasmacytoid DCs (pDCs) are a unique subset of DCs that can highly express transcription factor interferon regulatory factor 7 (IRF7) and intranuclear Toll-like receptor, enabling pDCs to rapidly secrete large amounts of IFN-α and interferon-β (IFN-β) to cope with viral infections [76]. Previous studies have found a large amount of pDCs infiltration in the dermis of VEGFA transgenic mice; pDCs have VEGFA receptors and migrate to VEGFA. Skin-infiltrating pDCs express VEGFR-1/Flt1 and VEGFR-2/Flk1, and VEGFA initiates psoriasis inflammatory cascade reaction by stimulating pDCs [77]. Similarly, CD11c+ DCs in VEGFA transgenic mice infiltrate more than in wild-type mice [78,79] (Figure 3).

### 4.4. VEGFA Regulates the Proliferation of Vascular Endothelial Cells

As mentioned earlier, pathological angiogenesis is one of the clinical features of psoriasis. Endothelial cells are the primary participating cells in the process of angiogenesis, forming the inner wall of blood vessels and usually referring to the single layer of flat epithelium on the inner surface of blood vessels and lymphatic vessels. Endothelial cells can engulf foreign bodies, bacteria, necrotic and aging tissues and can also participate in the immune activity functions of the body. In addition to secreting various cell adhesion factors, endothelial cells also express a set of specific receptors, namely tyrosine kinase receptors. Under normal circumstances, endothelial cells are quiescent, surrounded by pericytes, which inhibit their proliferation and secretion abilities. When hypoxia, inflammation, tumors, and other conditions occur, related cells release various pro-angiogenic factors, stimulating pericytes to separate, and the tight connection between them and endothelial cells fails, and new blood vessels begin to sprout [80,81]. In psoriasis, VEGFA expression regulates the biological processes of endothelial cells. In recent years, multiple studies have shown that tumor necrosis factor receptor-associated factor 3-interacting protein 2 (TRAF3IP2) is a strong candidate gene for psoriasis and psoriatic arthritis [82]. TRAF3IP2 is significantly upregulated in psoriatic skin lesions. Related to the VEGFA pathway, knocking down TRAF3IP2 can reduce the expression of VEGFA in human umbilical vein endothelial cells, block the cell cycle in the G2/M phase, and increase the expression of apoptotic proteins Caspase 3, Cleaved-Caspase 3, and Bax. TRAF3IP2 regulates the downstream VEGFA pathway, thereby affecting endothelial cell apoptosis. miR-205-5p can inhibit the proliferation, migration, and lumen formation of human umbilical vein endothelial cells. At the same time, miR-205-5p inhibits the proliferation of keratinocytes. VEGFA has been proven to be a target of miR-205-5p [83]. In psoriasis, VEGFA secreted by endothelial cells and keratinocytes binds to VEGFR-2/Flk1, mediating the activation of the downstream signaling pathway network of angiogenesis [84], as shown in Figure 4. It should be emphasized that VEGFR-2/Flk1 has a major role in VEGFA-induced angiogenesis, while VEGFR-1/Flt1 has an important role in the recruitment of hematopoietic precursor cells and the migration of monocytes. The affinity of VEGFA binding to VEGFR-1/Flt1 is higher than that of VEGFR-2/Flk1 [85]. However, the kinase activity of VEGFR-1/Flt1 is weak, and the binding of VEGFA on the surface of endothelial cells induces weak phosphorylation [86]. Both receptors are important in angiogenesis, which has been confirmed in some studies, i.e., it is necessary to block both VEGFR-1/Flt1 and VEGFR-2/Flk1 to prevent VEGFA-mediated angiogenesis.

### 4.5. VEGFA Inhibitors in Psoriasis

Currently, phototherapy, along with traditional and biological therapies used in clinical practice for treating psoriasis, all exhibit significant anti-angiogenic properties and can reduce the levels of VEGFA in the skin and plasma. The relevant data are shown in Table 2. This invisible mechanism of action undoubtedly contributes to its efficacy and should be further investigated. Anti-angiogenic therapies are being used increasingly in oncology and ophthalmology [87,88,89]; however, they have not yet been approved for psoriasis. Bevacizumab is the first humanized monoclonal antibody against tumor angiogenesis approved by the U.S. Food and Drug Administration, which can bind to VEGFA, reducing the binding of vascular endothelial growth factor to its receptor and thereby inhibiting vascular growth. A clinical study showed that when bevacizumab was used to treat a patient with metastatic renal cancer complicated with psoriasis and psoriatic arthritis (PsA), both psoriasis and PsA were completely alleviated without any other treatment. However, it was also found that his psoriasis flared up when he switched to other kinase inhibitors such as sorafenib or sunitinib, which suggested that bevacizumab may have promising prospects in the treatment of psoriasis and PsA. Previously, it was reported that a patient with metastatic colon cancer complicated with psoriasis experienced complete psoriasis remission without any other psoriasis treatment during the use of bevacizumab and combination chemotherapy. Of course, there is a general lack of clinical samples in current reports on psoriasis and VEGFA inhibitors in clinical research; nonetheless, it still reminds researchers to pay attention to the application of VEGFA inhibitors in psoriasis.

Given the development of this novel, targeted, personalized drug therapy, we advocate using VEGFA receptor inhibitors as adjunctive therapy to assist clinical biologics in accelerating the improvement of psoriasis severity scores [104,105,106]. In addition, for individuals with a significant “pro-angiogenic” psoriasis phenotype, targeted anti-angiogenic therapy can promote personalized treatment plans and personalized adjunctive therapy for well-defined subgroups of psoriasis patients, especially when they show incomplete responses to standard treatments. Considering the often (but certainly not always) impressive clinical improvements with several biologics commonly used in treating psoriasis, we do not advocate that anti-VEGF or VEGF receptor inhibition therapy should replace these biologics. Based on the existing evidence about the role of VEGFA in the pathogenesis of psoriasis and the clinical observations of patients who experienced psoriasis remission during anti-angiogenic treatment for other diseases, we believe it is necessary to systematically explore angiogenesis therapy for targeted treatment of psoriasis. Many adverse reactions have been associated with the systemic administration of VEGFA inhibitors, including proteinuria, hypertension, and impaired wound healing. Therefore, developing anti-VEGFA therapy for future psoriasis treatment needs to be carefully evaluated to minimize treatment-related toxicity. Relevant strategies to restore or reduce psoriasis vascular abnormalities may have significant therapeutic effects in patients with high levels of VEGFA.

## 5. Conclusions

Angiogenesis is tightly regulated at the molecular level in normal skin. However, dysregulation of angiogenesis occurs in various pathologies, leading to an abnormal vascular network characterized by dilated, tortuous, and hyperpermeable vessels [107]. Consequently, the concept of “vascular normalization” was proposed. Anti-angiogenic therapy is based on the premise of this hypothesis, aiming to restore the structure and function of blood vessels rather than simply eliminating them. Therefore, it may be necessary to combine anti-angiogenic drugs with other medications [108]. Significant vascular anomalies have been detected on the surface of psoriatic skin lesions. However, anti-angiogenesis therapy is not yet a mainstream treatment for psoriasis, and the currently available anti-angiogenic drugs still have certain side effects. For instance, bevacizumab may cause gastrointestinal reactions, bleeding, hypertension, proteinuria, and embolism [109,110]. Other anti-vascular drugs also exhibit varying degrees of side effects. The mechanism of action of these drugs indicates that while they exert therapeutic effects and inhibit abnormal angiogenesis, they may also harm normal blood vessels. Psoriasis patients often have comorbidities such as hypertension and metabolic syndrome, making them unsuitable candidates for anti-angiogenic therapy. Among the different forms of psoriasis, plaque psoriasis is most strongly associated with VEGFA. Therefore, testing VEGFA levels may be particularly important for patients with this type of psoriasis, while other forms may require more extensive clinical observation.

Regarding the role of angiogenesis in psoriasis, measuring VEGFA to define the angiogenic characteristics of psoriasis patients may serve as a predictive biomarker for the severity of psoriasis and the potential response to anti-angiogenic therapy. However, anti-VEGFA therapy is not currently a mainstream treatment for psoriasis. Despite a wealth of preclinical evidence about the role of VEGFA in the pathogenesis of psoriasis, its specific mechanism has not been thoroughly studied. In this perspective article, we detailed the relationship between VEGFA and psoriasis, where VEGFA mediates keratinocyte proliferation in psoriasis, regulates abnormal keratin expression in psoriasis plaques, regulates immune responses in the psoriasis epithelial immune microenvironment, and binds with downstream targets to promote endothelial cell proliferation. At the same time, it was found that therapeutic methods that reduce VEGFA expression can significantly alleviate psoriasis skin lesions. It is postulated that psoriasis patients with high VEGFA expression may be more prone to angiogenesis, making them susceptible to developing severe disease phenotypes. This unique group is most likely to benefit from anti-VEGFA treatment strategies, providing clinicians with opportunities for personalized treatment of psoriasis. Based on the information from this review research, we infer that patients with high levels of VEGFA may benefit the most from the anti-angiogenic therapy we advocate.

## Figures and Tables

**Figure 1 ijms-25-00059-f001:**
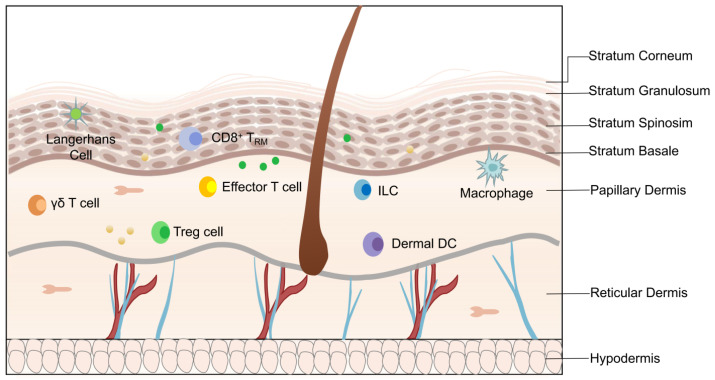
Structural and cellular components of skin. The structural components in epidermis: stratum corneum, stratum granulosum, stratum acanthosum, and stratum basalis. The structural components in dermis: papillary dermis and reticular dermis. Cell components in epidermis: keratinocytes, Langerhans cells, and CD8^+^ tissue-resident memory cells (CD8^+^ TRM). Cell components in dermis: γδ T cells, effector T cells, dermal dendritic cells (DCs), macrophages, and Treg cells.

**Figure 2 ijms-25-00059-f002:**
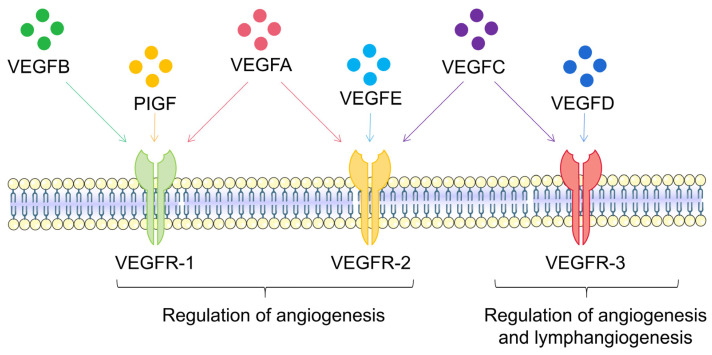
Schematic diagram of VEGF-VEGFR signaling pathway. The members of the VEGF family include VEGFA, VEGFB, VEGFC, VEGFD, VEGFE, and PlGF. VEGFA has an important role in regulating the normal body and disease-related angiogenesis. VEGFB, PIGF, and VEGFA combine with VEGFR-1; VEGFA, VEGFC, and VEGFE combine with VEGFR-2; VEGFC and VEGFD combine with VEGFR-3. The VEGFR-1 and VEGFR-2 systems are responsible for angiogenesis, while VEGFR-3 is responsible for the angiogenesis of both blood vessels and lymphatics.

**Figure 3 ijms-25-00059-f003:**
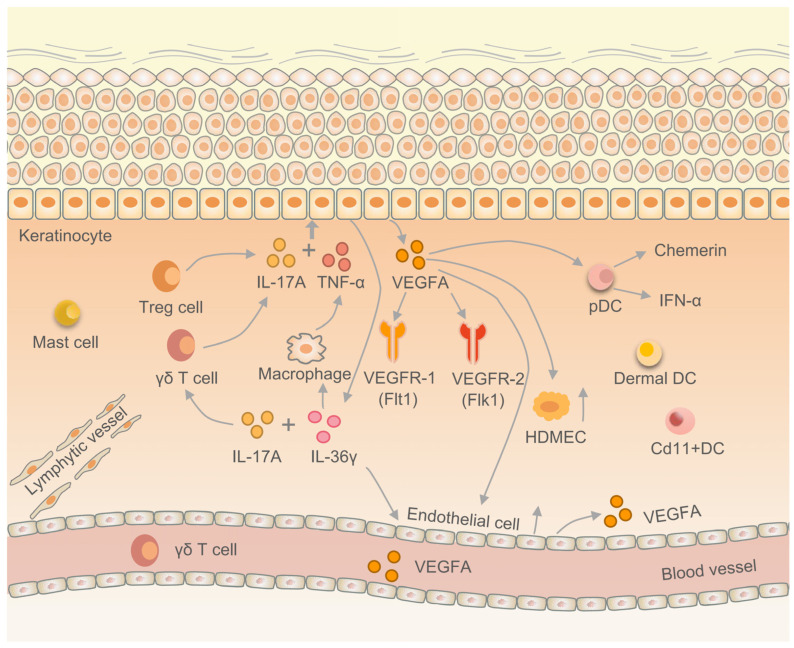
The role of VEGFA in the epithelial immune microenvironment of psoriasis. Th17 cell is a product of IL-17A. IL-17A associated with TNF-α stimulates keratinocytes to produce VEGFA. On the one hand, VEGFA induces pDCs to release chemerin and IFN-α, promoting the proliferation of HDMECs and endothelial cells. On the other hand, VEGFA, in combination with VEGFR1/Flt1 and VEGFR2/Flk1, activates downstream pathways.

**Figure 4 ijms-25-00059-f004:**
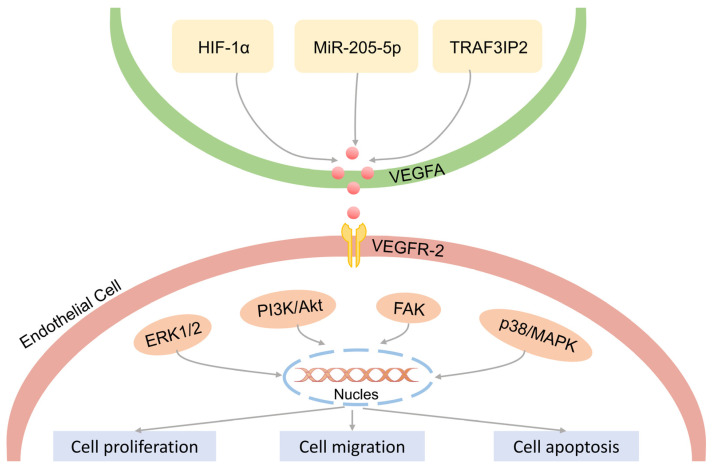
VEGFA regulates the biological behavior of vascular endothelial cells. Under the stimulation of the upstream signal of HIF-1α, MiR-205-5p, and TRAF3IP2, VEGFA further activates downstream pathways in vascular endothelial cells, such as ERK1/2, PI3K/Akt, FAK, and p38/MAPK, which further regulates the proliferation, migration, and apoptosis of vascular endothelial cells.

**Table 1 ijms-25-00059-t001:** Evidence for the role of VEGFA in psoriasis.

Research Stage	Discovery	Ref.
Preclinical research	Transgenic mouse models overexpressing VEGFA in keratinocytes lead to the main hallmarks of human psoriasis.	[48]
	VEGFA expression is enhanced in IL-17A-induced keratinocytes.	[49]
	In an imiquimod-induced psoriasis mouse model, the expression of VEGFA in skin lesions and serum is increased.	[50]
	SNPs of the *VEGF* gene, such as C at −460 to T in the promoter region and C at +405 to G in the 5′-untranslated region, confer a genetic predisposition associated with psoriasis risk.	[51,52]
Clinical research	The blood vessels in the papillary dermis of patients with psoriasis elongate, extend, and are tortuous. The expression of VEGFA and other pro-angiogenic growth factors in the skin lesions is increased.	[53,54]
	Anti-angiogenic therapy, e.g., bevacizumab and auranofin, may improve psoriatic lesions.	[55,56]

**Table 2 ijms-25-00059-t002:** VEGFA-associated therapies in the treatment of psoriasis in the last decade.

Type of Treatment	Drug	Findings	Ref.
Phototherapy	PUVA	Suppresses keratinocytes proliferation; Induces keratinocytes apoptosis; Regulates cytokine production.	[90]
	NB-UVB	Decrease the serum levels of VEGF and IL-8 in patients with psoriasis.	[91]
Standard Systemic Therapy	Retinoids	Normalization of keratinocyte proliferation/differentiation; Anti-proliferative effects.	[92]
	Methotrexate (MTX)	Reduces keratinocytes and lymphocyte proliferation; Induces lymphocyte apoptosis; Inhibits T-cell inflammatory action.	[90]
Biologic Therapy	Bevacizumab	Binds to VEGFA and its isoforms and inhibits interaction of VEGF with its receptors; Inhibition of new vessel formation and regression of existing vasculature.	[51,93]
	Adalimumab	Inhibition of endothelial cell proliferation; Narrowing of a vascular network; Reduction of vascular diameter.	[94]
	Midkine Monoclonal Antibody	Regulating VEGFA expression through the Notch Receptor 2 (Notch 2)/Hes Family bHLH Transcription Factor 1 (HES1)/Janus Kinase 2 (JAK2)-STAT5A pathway; Inhibition of keratinocyte proliferation; Reduces blood vessel density and inhibits angiogenesis.	[95]
Molecular Compounds	2-methoxyestradiol (2-ME)	2-ME induces apoptosis in keratinocytes; 2-ME blocked the G2/M phase and inhibited the proliferation of keratinocytes; 2-ME inhibits VEGFA induced by IL-17A; 2-ME alleviates psoriasis by inhibiting the JAK1/STAT3 pathway in vitro and in vivo.	[96]
	Thalidomide	Inhibition of keratinocyte activity; Decreased secretion of VEGF and TNF-a.	[97,98]
	Auranofin	Reduce mRNA expression levels of the VEGFA gene; Increased mRNA expression of IL-4 and IL-10 that was relevant to Th2/Treg cells; Reduce the mRNA expression levels of IFN-γ and IL-17A that were related to Th1/Th17 cells.	[56]
	Calcipotriol	LL-37 is implicated in the pathogenesis of psoriasis, induced angiogenesis; Inhibitors of scavenger receptors decrease the expression of VEGFA induced by LL-37 in keratinocytes.	[99]
	Astilbin	Astilbin could induce Nrf2 nucleus translocation; Reduce the ROS accumulation and VEGFA expression; Inhibit the proliferation of keratinocytes.	[100]
	Eosin	Dampens the release of pro-inflammatory chemokines (CCL2 and CCL5) and VEGFA.	[101]
	Ginsenoside Rh2	Inhibited VEGFA levels in the PN skin grafts.	[102]
	Gambogic acid	Inhibited proliferation of keratinocytes; Suppressed hyperplastic and inflamed vessels of *K14-VEGF* transgenic mice.	[103]

## Data Availability

No data were used for the research described in the article.

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
