# Peer review of "Vascular Endothelial Growth Factor A VEGFA Inhibition: An Effective Treatment Strategy for Psoriasis"

_ijms, 2023, doi:10.3390/ijms25010059_

Round 1

Reviewer 1 Report

Comments and Suggestions for Authors

The manuscript entitled ” VEGFA Inhibition: Effective Treatment Strategy for Psoriasis” by Ya Chen et al focuses on the uncovering the  role of blood vessels in psoriasis, explore the relationship between VEGFA and keratinocytes in the EIME.

The authors suggest VEGFA as a predictive biomarker for the severity of psoriasis and the potential response to anti-angiogenic therapy. 

The authors presented an interesting and informative review, which may be useful for other researchers. 

The following improvements in the article would help other researchers to understand the significance of the findings obtained in this work.

1. The suggestion of the authors based on this review, that patients with high levels of VEGFA may benefit the most from the anti-angiogenic therapy, looks logical. But the authors quite did not speak about negative aspects of anti-angiogenic therapy, and influence this therapy on the comorbitic diseases.

2. If the treatment of psoriasis by anti-angiogenic therapy is used, what is the novelty of the author's suggestion?

Comments on the Quality of English Language

 Minor editing of English language required

Author Response

Response to reviewer 1’s comments:

The manuscript entitled ” VEGFA Inhibition: Effective Treatment Strategy for Psoriasis” by Ya Chen et al focuses on the uncovering the role of blood vessels in psoriasis, explore the relationship between VEGFA and keratinocytes in the EIME.The authors suggest VEGFA as a predictive biomarker for the severity of psoriasis and the potential response to anti-angiogenic therapy.The authors presented an interesting and informative review, which may be useful for other researchers.The following improvements in the article would help other researchers to understand the significance of the findings obtained in this work.

Response: Thank you for the positive comments.

  1. The suggestion of the authors based on this review, that patients with high levels of VEGFA may benefit the most from the anti-angiogenic therapy, looks logical. But the authors quite did not speak about negative aspects of anti-angiogenic therapy, and influence this therapy on the comorbitic diseases.

Response: We appreciate and agree with the comments. We have added the negative aspects of anti-angiogenic therapy and the influence this therapy on the comorbitic diseases in the Discussion section. Please see page12 and13.

  1. If the treatment of psoriasis by anti-angiogenic therapy is used, what is the novelty of the author's suggestion?

Response: Thank you for the comments. Several mature anti-psoriatic drugs, including methotrexate [1], cyclosporine [2], and retinoids [3], as well as biologic therapies targeting immune factors involved in psoriasis pathogenesis, have been widely utilized in clinical practice [4]. While these therapies have demonstrated efficacy, their long-term effectiveness can be influenced by various factors. Challenges such as preventing recurrence, high cost, complex structure, and potential serious side effects need to be addressed [5]. Existing clinical data suggest that anti-angiogenic therapy appears to be a promising strategy for psoriasis treatment, with plaque psoriasis being the most extensively studied subtype [6-12]. Pharmacogenetic studies have revealed that psoriasis patients exhibit distinct genetic characteristics that correlate with treatment response and disease severity. A subset of psoriasis patients, characterized by high levels of VEGFA, may derive the greatest benefit from anti-VEGFA treatment. This presents an opportunity for personalized psoriasis treatment by assessing patients' VEGFA levels to identify those with elevated levels and tailor treatment accordingly. Such an approach facilitates the development of personalized treatment plans and enables precision medicine. Given that psoriasis primarily affects the skin surface, topical anti-angiogenic drugs could be developed to potentially mitigate unwanted side effects.

[1] Shaker, O. G., Khairallah, M., Rasheed, H. M., et al. (2013). Antiangiogenic effect of methotrexate and PUVA on psoriasis. Cell biochemistry and biophysics, 67(2), 735-742.

[2] Stinco, G., Lautieri, S., Valent, F., et al. (2007). Cutaneous vascular alterations in psoriatic patients treated with cyclosporine. Acta dermato-venereologica, 87(2), 152-154.

[3] Young, H. S., Summers, A. M., Bhushan, M., et al. (2004). Single-nucleotide polymorphisms of vascular endothelial growth factor in psoriasis of early onset. The Journal of investigative dermatology, 122(1), 209-215.

[4] Wang, J., Wang, Y. M., & Ahn, H. Y. (2014). Biological products for the treatment of psoriasis: therapeutic targets, pharmacodynamics and disease-drug-drug interaction implications. The AAPS journal, 16(5), 938-947.

[5] Rendon, A., & Schäkel, K. (2019). Psoriasis Pathogenesis and Treatment. International journal of molecular sciences, 20(6), 1475.

[6] Keshtgarpour M, Dudek AZ. (2007) SU‐011248, a vascular endothelial growth factor receptor‐tyrosine kinase inhibitor, controls chronic psoriasis. Transl Res, 149:103–6.

[7] Narayanan S, Callis‐Duffin K, Batten J et al. (2010) Improvement of psoriasis during sunitinib therapy for renal cell carcinoma. Am J Med Sci, 339:580–1.

[8] Fournier C, Tisman G. (2010) Sorafenib‐associated remission of psoriasis in hypernephroma: case report. Dermatol Online J, 16:17.

[9] Antoniou EA, Koutsounas I, Damaskos C, et al. (2016) Remission of psoriasis in a patient with hepatocellular carcinoma treated with sorafenib. In Vivo, 30:677–80.

[10] van Kester MS, Luelmo SAC, Vermeer MH et al. (2018) Remission of psoriasis during treatment with sorafenib. JAAD Case Rep, 4:1065–7.

[11] Akman A, Yilmaz E, Mutlu H et al. (2009) Complete remission of psoriasis following bevacizumab therapy for colon cancer. Clin Exp Dermatol, 34:e202–4.

[12] Datta‐Mitra A, Riar NK, Raychaudhuri SP. (2014) Remission of psoriasis and psoriatic arthritis during bevacizumab therapy for renal cell cancer. Indian J Dermatol, 59:632.

Reviewer 2 Report

Comments and Suggestions for Authors

The manuscript titled “VEGFA Inhibition: Effective Treatment Strategy for Psoriasis” IJMS (ISSN 1422-0067) presents an interesting review of the relationship between VEGFA and keratinocytes in the epithelial immune microenvironment – EIME. The review is very interesting and gives a new inside into the role of VEGFA in psoriasis.

1.       There is various type of psoriasis. What kind of psoriasis correlates the most with the concertation of VEGFA?

2.       Is there a proven correlation between VEGFA and the severity of psoriasis in clinical studies?

3.       If VEGFA can be measured in clinical practice, which patients should have this parameter measured? All patients with psoriasis or patients with specific types of psoriasis?

4.       Psoriasis is a component of psoriatic arthritis (PsA). Is VEGFA concentration also changed in PsA?

5.       Hypothetically, if an anti-VEGFA molecule is available for treatment, which patients (what kind of psoriasis) will benefit most from it?

Author Response

Response to reviewer 2’s comments:

The manuscript titled “VEGFA Inhibition: Effective Treatment Strategy for Psoriasis” IJMS (ISSN 1422-0067) presents an interesting review of the relationship between VEGFA and keratinocytes in the epithelial immune microenvironment–EIME. The review is very interesting and gives a new inside into the role of VEGFA in psoriasis.

Response: Thank you for the positive comments.

  1. There is various type of psoriasis. What kind of psoriasis correlates the most with the concertation of VEGFA?

Response: Thank you for the comments. Based on the existing literature on psoriasis and VEGFA, vulgaris psoriasis is the most commonly reported subtype. A study conducted on patients with chronic plaque psoriasis found a significant increase in the expression of VEGF and its receptors compared to a healthy control group. This suggests that alterations in VEGF biology may contribute to the development of plaque psoriasis [1]. Additionally, the study revealed a significant association between the "high VEGFA producing" genotype and early-onset plaque psoriasis as well as the development of severe disease [2]. VEGFA-mediated angiogenesis has been identified as a contributing factor in the formation of psoriatic plaques [3]. Furthermore, seven patients with vulgaris psoriasis experienced improvement after receiving treatment for other conditions using antiangiogenic drugs that target the VEGFA/VEFR pathway [4-10]. As per your suggestion, we have included the relevant information on page 13.

  • Young, H. S., Summers, A. M., Bhushan, M. et al. (2004). Single-nucleotide polymorphisms of vascular endothelial growth factor in psoriasis of early onset. The Journal of investigative dermatology, 122(1), 209–215.
  • [2] Young, H. S., Summers, A. M., Read, I. R. et al. (2006). Interaction between genetic control of vascular endothelial growth factor production and retinoid responsiveness in psoriasis. The Journal of investigative dermatology, 126(2), 453-459.

[3] Luengas-Martinez, A., Paus, R., & Young, H. S. (2022). Antivascular endothelial growth factor-A therapy: a novel personalized treatment approach for psoriasis. The British journal of dermatology, 186(5), 782–791.

[4] Keshtgarpour M, Dudek AZ. (2007) SU-011248, a vascular endothelial growth factor receptor-tyrosine kinase inhibitor, controls chronic psoriasis. Transl Res, 149:103-6.

[5] Narayanan S, Callis-Duffin K, Batten J et al. (2010) Improvement of psoriasis during sunitinib therapy for renal cell carcinoma. Am J Med Sci, 339:580-1.

[6] Fournier C, Tisman G. (2010) Sorafenib-associated remission of psoriasis in hypernephroma: case report. Dermatol Online J, 16:17.

[7] Antoniou EA, Koutsounas I, Damaskos C et al. (2016) Remission of psoriasis in a patient with hepatocellular carcinoma treated with sorafenib. In Vivo, 30:677-80.

[8] van Kester MS, Luelmo SAC, Vermeer MH et al. (2018) Remission of psoriasis during treatment with sorafenib. JAAD Case Rep, 4:1065-7.

[9] Akman A, Yilmaz E, Mutlu H et al. (2009) Complete remission of psoriasis following bevacizumab therapy for colon cancer. Clin Exp Dermatol, 34:e202-4.

[10] Datta-Mitra A, Riar NK, Raychaudhuri SP. (2014) Remission of psoriasis and psoriatic arthritis during bevacizumab therapy for renal cell cancer. Indian J Dermatol, 59:632.

  1. Is there a proven correlation between VEGFA and the severity of psoriasis in clinical studies?

Response: Thank you for the comments. Extensive clinical studies have indeed demonstrated a clear correlation between VEGFA and the severity of psoriasis. These studies have revealed that serum VEGFA levels decrease during psoriasis remission [1-6]. Moreover, the expression of VEGFA in the skin tissue of psoriasis patients is significantly higher compared to normal skin tissue. Notably, the level of VEGFA in the skin tissue exhibits a strong positive correlation with the clinical severity of psoriasis, as evidenced by an impressive R-value of 0.9 [7].

  • Andrys, C., Borska, L., Pohl, D. et al. (2007). Angiogenic activity in patients with psoriasis is significantly decreased by Goeckerman's therapy. Archives of dermatological research, 298(10), 479-483.
  • Shaker, O. G., Khairallah, M., Rasheed, H. M. et al. (2013). Antiangiogenic effect of methotrexate and PUVA on psoriasis. Cell biochemistry and biophysics, 67(2), 735-742.
  • Wen, J., Wang, X., Pei, H. et al. (2015). Anti-psoriatic effects of Honokiol through the inhibition of NF-κB and VEGFR-2 in animal model of K14-VEGF transgenic mouse. Journal of pharmacological sciences, 128(3), 116-124.
  • Deng, H., Yan, C. L., Hu, Y. et al. (2004). Photochemotherapy inhibits angiogenesis and induces apoptosis of endothelial cells in vitro. Photodermatology, photoimmunology & photomedicine, 20(4), 191-199.
  • Yamasaki, E., Soma, Y., Kawa, Y. et al. (2003). Methotrexate inhibits proliferation and regulation of the expression of intercellular adhesion molecule-1 and vascular cell adhesion molecule-1 by cultured human umbilical vein endothelial cells. The British journal of dermatology, 149(1), 30-38.
  • Johnston, A., Gudjonsson, J. E., Sigmundsdottir, H. et al. (2005). The anti-inflammatory action of methotrexate is not mediated by lymphocyte apoptosis, but by the suppression of activation and adhesion molecules. Clinical immunology (Orlando, Fla.), 114(2), 154-163.
  • Bhushan M, McLaughlin B, Weiss JB et al. (1999) Levels of endothelial cell stimulating angiogenesis factor and vascular endothelial growth factor are elevated in psoriasis. Br J Dermatol, 141:1054–60.

  1. If VEGFA can be measured in clinical practice, which patients should have this parameter measured? All patients with psoriasis or patients with specific types of psoriasis?

Response: Thank you for the comments. Patients with vulgaris psoriasis [1-7] may benefit from having their VEGFA levels measured. However, it is important to note that different types of psoriasis may require different approaches to clinical observation. Additionally, it is worth considering that psoriasis patients often have comorbidities such as hypertension and metabolic syndrome, which are commonly associated with cardiometabolic diseases. As anti-angiogenic therapy can lead to adverse reactions such as hypertension and proteinuria, it is generally not recommended for psoriasis patients with cardiometabolic diseases. We appreciate your suggestion and have included the relevant information on page 13.

[1] Keshtgarpour M, Dudek AZ. (2007) SU-011248, a vascular endothelial growth factor receptor-tyrosine kinase inhibitor, controls chronic psoriasis. Transl Res, 149:103-6.

[2] Narayanan S, Callis-Duffin K, Batten J et al. (2010) Improvement of psoriasis during sunitinib therapy for renal cell carcinoma. Am J Med Sci, 339:580-1.

[3] Fournier C, Tisman G. (2010) Sorafenib-associated remission of psoriasis in hypernephroma: case report. Dermatol Online J, 16:17.

[4] Antoniou EA, Koutsounas I, Damaskos C et al. (2016) Remission of psoriasis in a patient with hepatocellular carcinoma treated with sorafenib. In Vivo, 30:677-80.

[5] van Kester MS, Luelmo SAC, Vermeer MH et al. (2018) Remission of psoriasis during treatment with sorafenib. JAAD Case Rep, 4:1065-7.

[6] Akman A, Yilmaz E, Mutlu H et al. (2009) Complete remission of psoriasis following bevacizumab therapy for colon cancer. Clin Exp Dermatol, 34:e202-4.

[7] Datta-Mitra A, Riar NK, Raychaudhuri SP. (2014) Remission of psoriasis and psoriatic arthritis during bevacizumab therapy for renal cell cancer. Indian J Dermatol, 59:632.

  1. Psoriasis is a component of psoriatic arthritis (PsA). Is VEGFA concentration also changed in PsA?

Response: Thank you for the comments. Psoriatic arthritis (PsA) is a systemic disease that affects the joints and is often seen in patients with psoriasis [1]. We conducted a search on PubMed using "PsA" and "VEGFA" as keywords and found a limited number of articles discussing the relationship between them. These studies used immunohistochemical and in situ hybridization probes to examine levels of the angiogenic growth factor VEGF in individuals with psoriatic arthritis compared to a control group. The results showed a significant increase in VEGF expression in the early stages of PsA, indicating a strong correlation between PsA and VEGFA concentration [2]. Another study involving 9 patients with psoriatic arthritis who had active polyarthritis despite methotrexate treatment found that the clinical efficacy of infliximab, a medication used to treat PsA, correlated with changes in factors related to angiogenesis regulation. Specifically, VEGF levels decreased while Ang-2 levels increased, suggesting that vascular degeneration may be a potential mechanism for the anti-angiogenic effects of infliximab [3].

[1] Eder, L., Haddad, A., Rosen, C. F. et al. (2016) The Incidence and Risk Factors for Psoriatic Arthritis in Patients With Psoriasis: A Prospective Cohort Study. Arthritis & rheumatology (Hoboken, N.J.), 68(4), 915–923.

[2] Fearon, U., Griosios, K., Fraser, A. et al. (2003) Angiopoietins, growth factors, and vascular morphology in early arthritis. The Journal of rheumatology, 30(2), 260-268.

[3] Cañete, J. D., Pablos, J. L., Sanmartí, R. et al. (2004) Antiangiogenic effects of anti-tumor necrosis factor alpha therapy with infliximab in psoriatic arthritis. Arthritis and rheumatism, 50(5), 1636-1641.

  1. Hypothetically, if an anti-VEGFA molecule is available for treatment, which patients (what kind of psoriasis) will benefit most from it?

Response: Thank you for the comments. Based on current clinical and basic research data, it is suggested that psoriasis vulgaris, especially the type associated with plaque formation, may benefit the most from molecular therapy targeting VEGFA. Studies have shown that VEGFA levels are significantly increased in plaques of psoriasis compared to normal skin [1], and these levels seem to be correlated with the severity of the disease [2]. Furthermore, the use of the VEGFA monoclonal antibody bevacizumab (Avastin®) has shown improvement in plaque psoriasis [3, 4].

[1] Detmar M, Brown LF, Claffey KP et al. (1994) Overexpression of vascular permeability factor/vascular endothelial growth factor and its receptors in psoriasis. J Exp Med, 180:1141–6.

[2] Bhushan M, McLaughlin B, Weiss JB, et al. (1999) Levels of endothelial cell stimulating angiogenesis factor and vascular endothelial growth factor are elevated in psoriasis. Br J Dermatol, 141:1054–60.

[3] Akman A, Yilmaz E, Mutlu H et al. (2009) Complete remission of psoriasis following bevacizumab therapy for colon cancer. Clin Exp Dermatol, 34:e202-4.

[4] Datta-Mitra A, Riar NK, Raychaudhuri SP. (2014) Remission of psoriasis and psoriatic arthritis during bevacizumab therapy for renal cell cancer. Indian J Dermatol, 59:632.